# Factors Influencing the Intention to Use the Common Ticketing System (Spider Card) in Thailand

**DOI:** 10.3390/bs9050046

**Published:** 2019-04-28

**Authors:** Fasang Prayoonphan, Xiaolin Xu

**Affiliations:** College of Public Administration, Huazhong University of Science and Technology, Wuhan 430074, China; xiaolin@hust.edu.cn

**Keywords:** the common ticket, Spider Card, Thailand, intention to use, UTAUT, perceived value

## Abstract

The Common Ticket, locally called the “Spider Card” is a contactless smartcard ticket used for public transportation systems in Thailand. The card is used for all modes of transportation and increases the convenience of public transit passengers while increasing public transport ridership. This research aimed to identify the determinant factors that impact passengers’ intention to use the Spider Card based on the unified theory of acceptance and use of technology (UTAUT) model by integrating perceived convenience and perceived sacrifice as part of the perceived value. A survey of public transit users in the Bangkok metropolitan area was conducted. The Partial Least Square-Structure Equation Modelling (PLS-SEM) technique was employed to examine the data. The results showed that perceived value, performance expectancy, and facilitating conditions are all significant factors. Perceived convenience and perceived sacrifice are significant antecedents for perceived value. Surprisingly, effort expectancy and social influence did not impact passengers’ intentions to use the Spider Card. The results also offer beneficial information for public transit authorities in order to comprehend what passengers desire from using these kinds of technology service systems.

## 1. Introduction 

The technology of contactless smartcards has been adopted by transit agencies around the world for transit fare collection since 1990 [1,2]. Many cities have implemented this kind of technology scheme, making it integral to modern public transport systems [3]. Contactless smartcards offer opportunities for transport agencies to coordinate fare collections with other agencies, improve data collection about how riders use transit, and reduce fare fraud [4]. Moreover, passengers can make a journey that involves transfers within or between different transport modes through an all-in-one ticket [5]. This type of card offers advantages and makes it more convenient when compared to more traditional payment methods [6]. Integrated ticketing technology is a kind of contactless smartcard for a fare collection payment system used in public transport systems worldwide. Turner and Wilson [7], as well as other researchers, have explained the concept of integrated ticketing in that it can be paper or smart tickets that are suitable for use in multiple modes of transport. The advantages of these systems are provided to both public transit passengers and transport operators. Focusing on Asian countries, EZ-Link in Singapore, the Octopus card in Hong Kong or Touch ‘n Go in Malaysia are good examples of the success of adopting an integrated smartcard ticketing system for public transit services [8]. 

Thailand recently made the first integrated smartcard ticketing available for use in multi-modal public transit in mid-June of 2018, called the “Spider Card” and locally named the MANG-MOOM card (Thai for “Spider”) [9,10]. This multi-modal smartcard payment scheme is still a new service and was first officially launched in Thailand for the early stage. In order to achieve wider acceptance for using the integrated smartcard ticketing system in Thailand, this study aims to identify the dominant factors that influence passengers’ intention to use the Spider Card for public transportation in Thailand. The intention to use the integrated smartcard ticket is affected by many factors including performance, ease of use, and personal expectations, as well as the infrastructure facilitating the passengers, social impact, pricing, value perception and convenience, and travel time savings when using the Spider Card. More complete understanding of user intention and adoption of the integrated ticketing system is still needed as there have been limited published studies focused on the factors which might influence integrated ticketing usage behaviour. Moreover, researchers in the past five years have shown significant interest in mobile-payment adoption rather than smartcard ticketing adoption [6,11]. Several researchers have applied the unified theory of acceptance and use of technology (UTAUT) in various contexts [12,13,14,15]. Therefore, UTAUT could be appropriate for employment in this study. Prior studies have been carried out by utilising the role of perceived value, which is widely employed in the marketing industry to estimate the public transit passengers’ intention behaviour. The legitimacy of perceived value has also been assessed experientially [16,17,18]. Other researchers have developed their research framework by incorporating a perceived value construct into the determinant of UTAUT constructs in order to explore the intention of using information system technology [19,20]. This study is quantitative research. The results from this study will be used to make useful suggestions for government agencies and the transport policy authority in order to build a better understanding of passengers’ intention to use the Spider Card and ways to increase the ridership on public transit services and its usage.

### 1.1. Theoretical Foundation

#### 1.1.1. Unified Theory of Acceptance and Use of Technology (UTAUT) and Its Dimensions

Several technology adoption theories have often been used for research studies in order to forecast personal different intention to adopt new technologies. A Unified Theory of Acceptance and Use of Technology (UTAUT) is a unified technology acceptance models formulated by Venkatesh, et al. [21], as a conceptual framework for understanding users’ intention and acceptance of technology in several contexts [22,23]. The UTAUT model integrates eight dominant models of acceptance and use of technology theories including the Theory of Reason Action (TRA) proposed by Fishbein and Ajzen [24], the Technology Acceptance Model (TAM) proposed by Davis [25] and Davis, et al. [26], the Theory of Planned Behavior (TPB) proposed by Ajzen [27], Combined TAM and TPB (C-TAM-TPB) proposed by Taylor and Todd [28], the Motivation Model (MM) proposed by Vallerand [29], the Model of PC Utilization (MPCU) proposed by Triandis [30], the Diffusion of Innovation Theory (DOI) proposed by Rogers [31], and Social Cognitive Theory (SCT) proposed by Bandura [32], which is connected to all eight theories above as the unified theory between the main factors and latent variables. The key determinant constructs affect the intended use of new technology perspectives, which can be determined by three constructs including performance expectancy, effort expectancy and social influence. Performance expectancy is defined as the degree to which an individual believes that using the system will help them to accomplish gains in job performance [21], while effort expectancy can be defined as the degree of ease associated with consumers’ use of technology which explains an individual’s perception of the level of difficulty or ease of using an IT system [21] as well as a being a predictor for an individual’s intention to use IT [33]. Social influence is the extent to which individuals perceive the degree of a certain behavior by important referents (e.g., family and friends) that they should use a particular technology [22]. Moreover, Venkatesh, et al. [21] indicated that facilitating conditions and behavioral intention also have a direct effect on usage behavior. Facilitating conditions are defined as the degree to which an individual believes that an organizational and technical infrastructure exists to support use of the system [22]. Also, there are four individual characteristics including gender, age, experience, and voluntariness of use as moderators of the UTAUT model [21], However, the moderating variables were not mentioned in this study.

The UTAUT has been used to explain behavioral intention and used in various contexts of information technology such as e/m-government services [33,34], e/m-payment [11,19,35,36], mobile technology [20], and transportation systems [13,37]. Previous studies have applied the UTAUT model to explore the context of transit fare payment technology adoption, as seen in Wu, et al. [37] where the UTAUT model was adopted to investigate citizens’ acceptance and use of iPass transit smartcard in Taiwan’s Kaohsiung MRT system. The results showed that most constructs of the UTAUT have a strong positive influence on behavioural intention. This study also takes facilitating conditions as one of the determinant factors affecting the intention to use the Spider Card. Previous studies have proven that facilitating conditions have a positive influence on technology adoption and intention [22]. 

#### 1.1.2. The Role of Perceived Value

Perceived value is the major element of any successful marketing strategies. As first offered by Zeithaml [38], perceived value refers to an overall assessment of a product or service from the consumers based on perceptions of what is received and what is given. Numerous scholars have extended their understanding of perceived value as the ratio or trade-off between benefit and cost, e.g., [18,39,40,41], which is a value-for-money concept. In general, perceived value can be linked to the value that consumers believe they will get from buying certain products and services. This is done by comparing with other similar products and services in terms of the financial and non-financial aspects of purchased products and services [38,42]. Zeithaml [38] argued that some people view value as meaning a low price, while others see it as relative equality between the quality and price of a purchase. Another researcher, Lovelock [42] also suggested that perceived value may be considered a compromise between perceived benefits and perceived cost. Additionally, perceived value will probably be high when the perceived benefit is greater than any perceived cost. The opposite is also true [43].

A commonly accepted influence factor on behavioural intention and adoption is perceived value. A number of studies [16,18,43,44] revealed that perceived value could be impacted by various precursor variables. Focusing on the transportation context, Lai and Chen [16] examined the influence of perceived value on behavioural intentions for major transport facilities. Apparently, perceived value has a direct influence on behavioural intentions, as found in their study. Sumaedi, et al. [45] likewise identified the importance of perceived value for the intention of users of public transit in Jakarta, Indonesia. Therefore, it could be assumed that perceived value is a significant element of behavioural intentions to use common ticketing among passengers. Thus, the intention to use a common ticketing system will be high if passengers perceive high value.

#### 1.1.3. Perceived Convenience

Convenience is typically considered and debated in marketing and consumer behaviour literature [46]. Convenience is also recognised as an aspect of consumers’ attitudes in terms of the time and effort needed for buying specific goods and services [46]. Yoon and Kim [47] described perceived convenience as the degree of convenience individuals realise when using services to complete a task measured by time, location and accomplishment. Moreover, Colwell, et al. [48] revealed that service convenience can be classified as a way to create added value for consumers by reducing the time and effort that they have to spend for products and services. When a product or service saves time for a consumer, it is deemed convenient. Conversely, a product or service is viewed as being convenient if it decreases the mental, emotive and physical drain felt by consumers. Based on the perspective provided by [46,47], in the context of transport services in this research, perceived convenience was defined as the degree of passengers’ recognition of convenience due to using the Spider Card as measured by less travel time and ease of inter-modality transport, in addition to time savings in comparison with conventional pay systems. Further, experiential research by Liu, et al. [49] likewise showed that perceived convenience has an affirmative impact on perceived value toward the behavioural intention of using mobile coupon applications.

#### 1.1.4. Perceived Sacrifice

Perceived sacrifice typically concerns what must be surrendered or paid in order to gain certain goods or services [38]. As seen in the literature from previous studies, perceived sacrifice may be separated into monetary and non-monetary, such as time, search, and psychological costs [38,43,50]. A sacrifice is obvious if a consumer must use time and travel some distance to buy a product or service. Moreover, on the basis of Zeithaml’s (1988) work, perceived sacrifice is often revealed as an indicator of the perceived value. In previous works relating to transportation, an immediate and adverse bond between perceived sacrifice and perceived value is verifiable. For example, the study conducted by Wen, et al. [43] revealed that perceived sacrifice is noteworthy according to statistics, meaning it has a direct unfavourable impact on perceived value. This indicates that low perceived value could be a consequence of high perceived sacrifice. Quite the opposite, low perceived sacrifice could bring about high perceived value. 

### 1.2. Conceptual Framework and Hypotheses

#### 1.2.1. Performance Expectancy

One key construct from the UTAUT model is performance expectancy (PE) [21]. Performance expectancy is one of the most meaningful elements that can promptly affect behavioural intentions under many conditions according to past studies [12,13,14,33], including an empirical study in the context of an E-ticketing system [33]. Therefore, in this study, performance expectancy means users’ belief that using the Spider Card can offer benefits in daily life and increase relative advantages more than using traditional tickets. Thus, the study hypothesises that:
**Hypothesis** **1** **(H1):**Performance expectancy will have a positive effect on intention to use Spider Card.

#### 1.2.2. Effort Expectancy

A primary construct from UTAUT that can be deemed as the level of ease consumers should feel when using technology is called effort expectancy (EE), which can be utilised to support individuals’ awareness of the degree of struggle or comfort felt when using an IT system [18]. Previous research conducted by [13,20,21,37] have found that effort expectancy has a direct and significant effect on behavioural intentions. For this study, effort expectancy is explained as the belief among users that the Spider Card is easy to use and there are no difficulties identified in using this system. Assuming that users feel that they do not encounter any obstacles while using the Spider Card and that it is convenient, users will probably have higher expectations for achieving the expected performance [36]. Moreover, Herrero, et al. [15] also proved that effort expectancy is positively related to performance expectancy for the adoption of social network sites used in the tourism industry in Spain. Thus, the study hypothesises that:
**Hypothesis** **2** **(H2):**Effort expectancy will have a positive effect on intention to use Spider Card.
**Hypothesis** **3** **(H3):**Effort expectancy will have a positive effect on performance expectancy.

#### 1.2.3. Facilitating Conditions

Based on UTAUT, facilitating conditions (FC) obviously affect usage behaviour [21]. However, previous studies have demonstrated a link with behavioural intentions in many settings. Bhuasiri, et al. [33] revealed that facilitating conditions are significant for impacting taxpayers’ incentive to use the e-tax filing and payment system. In this study, facilitating conditions describe a variety of factors that are able to facilitate the use of the Spider Card such as the location and way to top-up money, facility support from card operators and relevant agencies, and assistance available from ground staff as well as other things that are needed when using the Spider Card. Thus, the study hypothesises that:
**Hypothesis** **4** **(H4):**Facilitating conditions will have a positive effect on intention to use Spider Card.

#### 1.2.4. Social Influence

How much a person believes that it is vital for other people to consider that the individual should use new technology is called social influence (SI) [21]. Social influence in the current research is defined as societal stress by family and known associates who can influence the intent to use the Spider Card. Users also perceive that using the Spider Card can enhance their personal image. Previous studies have shown that social influence is a critical indicator of behavioural intention for technology adoption [19,33,51]. Empirical evidence from Wu, et al. [37] demonstrated that social influence has a significant and positive influence on the behavioural intention to use the iPass smartcard, which is used by the Kaohsiung MRT system. Thus, the study hypothesises that:
**Hypothesis** **5** **(H5):**Social influence will have a positive effect on intention to use Spider Card.

#### 1.2.5. Perceived Convenience

Consumers’ opinions relating to time and effort can be used to explain the concept of convenience in terms of buying or using goods and services [46]. Meanwhile, convenience is often detailed as several dimensions including time, place, and execution, as suggested by previous studies concerning technological innovation [47,49]. Liu, et al. [49] defines perceived convenience (PC) as consumer’s opinion that technology can be used to carry out tasks with more convenience in terms of time savings, location, and manner. Based on the perspective provided by the definition as mentioned above of perceived convenience, in the present study it is defined as passengers’ recognition of the convenience gained from using the Spider Card in the form of reduced travel time, convenience for inter-modality, time savings when compared with using tokens, coins or cash. Moreover, an empirical study from Liu, et al. [49] indicated that perceived convenience has positive effects on perceived value for the behavioural intention of using mobile coupon applications. Thus, this study hypothesises that:
**Hypothesis** **6** **(H6):**Perceived convenience will have a positive effect on perceived value.

#### 1.2.6. Perceived Sacrifice

What has been surrendered to get a product or service is called perceived sacrifice (PS) [38]. Perceived sacrifice may be categorised as monetary and non-monetary, such as time, search, and psychological costs, as detailed in previous studies. Thus, monetary prices in this study refer to the actual fare prices that passengers must pay if they use the Spider Card, while non-monetary prices refer to the time associated with usage. Furthermore, previous studies have proven that the effect of sacrifice aspects are negatively related to perceived value. The implication is that low perceived value could come from high perceived sacrifice. In the same way, low perceived sacrifice could cause high perceived value [43]. In the literature concerning passengers’ behavioural intention, it is demonstrated that perceived value is significantly affected by perceived sacrifice. Wen, et al. [43] found that perceived sacrifice was likely an indicator of perceived value. Similarly, Sumaedi, et al. [45] showed that perceived sacrifice is measurably important and has a direct, adverse impact on perceived value. Thus, the connection between perceived sacrifice and perceived value is noteworthy in this work. Thus, this study hypothesises that:
**Hypothesis** **7** **(H7):**Perceived sacrifice will have a negative effect on perceived value.

#### 1.2.7. Perceived Value

In research concerning technology adoption, perceived value (PV) is often employed to clarify passengers’ motivation and use in a variety of circumstances. Previous studies have found that perceived value may serve as an indicator of behavioural intention [44]. Lai and Chen [16] applied the perceived value construct to explore the relationship between passenger behavioural intentions in the public transportation context. Their research showed that perceived value has a significant, positive influence on behavioural intention. This is consistent with the recent study of Irtema, et al. [17], which examined the behavioural intention of public passengers in the capital city of Malaysia. The results indicated that perceived value had an effective impact on behavioural intention according to the existing literature and based on a study conducted by Lai and Chen [16]. Therefore, the study hypothesises that:
**Hypothesis** **8** **(H8):**Perceived value will have a positive effect on intention to use the Spider Card.

### 1.3. Proposed Research Model

On the basis of the literature review and hypotheses presented above, the proposed research model is presented in Figure 1. This model illustrates the certain hypothesized relationships among the intention to use, the construct of the UTAUT model and the value perceptions aspect. 

## 2. Materials and Methods

### 2.1. Measurement Instrument

Using the elements of the proposed model, all calculation items were designed to measure with a five-point Likert scale ranging from strongly disagree (1) and strongly agree (5). Existing literature was used to choose the constructs of the model, which were then adapted to reflect the transport users’ motivation to use the Spider Card in Thailand. Numerous items, primarily modified from instruments confirmed in previous studies were used to measure the eight constructs. These constructs were changed to conform to the context of this study. Items of performance expectancy, facilitating conditions and social influence were measured by the scale from Venkatesh, et al. [21], Wu, et al. [37] and Yeow and Loo [52]. Items of effort expectancy were developed from Venkatesh, et al. [21], Wu, et al. [37] and Davis [25]. Items of perceived convenience were modified from Yoon and Kim [47] and Liu, et al. [49]. Items of perceived sacrifice were adapted from Jen and Hu [18] and Wen, et al. [43]. Items of perceived value were revised from Liu, et al. [49] and items of intention to use the Spider Card were measured by scales derived from Venkatesh, et al. [21] and Wu, et al. [37]. The final scales and items are presented in Appendix A.

### 2.2. Samples and Data Collection

The Spider Card was first officially available for public transportation in the capital city of Thailand. Therefore, the people living, working, studying, or doing business in the Bangkok metropolitan area with access to public transportation were employed in this study. In order to prevent disturbing the respondents who use public transport, the research data was collected through an online survey created by Google Forms. Respondents could complete a survey at their convenience using the relatively autonomous online approach. Additionally, past research reinforces the suitability of using an online approach to carry out a survey more cost effectively and with time savings [49]. The online questionnaire was created using Google Forms and distributed from 13 October 2018 to 30 November 2018 via social network sites including Line application, Facebook, WeChat, and Twitter. People living, working, studying or doing business and having experience with public transportation usage in the Bangkok metropolitan area were invited to participate in the survey. Overall, 408 respondents completed the online questionnaire, with 20 of them being excluded because of the fallibility of answers. Three hundred-eighty-eight responses at a response rate of roughly 95 per cent comprised the final dataset. According to Cohen’s rule [53], a robust effect size for the sample size should be at least 10 times the amount of the hypotheses. The research model examined 8 hypotheses; therefore, a sample size of at least 80 is suggested. In this research, a final dataset of 388 fulfils this recommendation and is appropriate for appraising the links in the hypotheses. Among the participants, 70.4 percent were female and 29.6 percent were male. 81.44 percent of participants were between 25 and 45 years old. More than 97.16 percent possessed a bachelor’s degree education or higher, while 78.61 percent were government officers and company employees. More than 97.68 percent used the Spider Card less than 3–5 times per week. A detailed description of respondent characteristics is shown in Table 1. 

### 2.3. Data Analysis Technique

This study employed the Partial Least Square-Structural Equation Modelling (PLS-SEM) technique performed on SmartPLS Version 2.0 (SmartPLS GmbH, Bönningstedt, Germany) [54] to test the proposed model hypotheses. Because of the small sample size and because the study assesses latent variables, the PLS-SEM approach is suitable for data analysis. The two-step method suggested by Chin [55] was used to assess the research model. Initially, the measurement model was examined to verify the dependability and legitimacy of the constructs. This was followed by checking the structural model to investigate the hypotheses. The evaluation of PLS-SEM and output reporting followed the approach outlined by Hair Jr, et al. [56].

## 3. Results

### 3.1. Measurement Model

Based on previous studies, the questionnaire items were established in English by the researcher. The phasing of the questions was modified according to the pre-test results and suggestions from transport and traffic policy analysts under Thailand’s Ministry of Transport. Afterwards, a linguistics specialist translated the questionnaire into Thai. Prior to conducting the survey, a pre-test of the questionnaire was done. Cronbach’s alpha was used to test the consistency of the questionnaire and consequently verify its contents [57]. Internal dependability was established using Cronbach’s alpha and Composite Reliability (CR) at a value of 0.70 or greater in order to evaluate the construct reliability.

Convergent validity is appraised utilising the CR and the Average Variance Extracted (AVE) gained from the valuation. The values for Cronbach’s alpha and CR in all concepts were higher than the 0.7 level suggested by Henseler, et al. [58], signifying the high dependability of the scales. Simultaneously, the AVE value for all concepts were beyond the 0.5 ceiling suggested by Hair Jr, et al. [56]. As seen from Table 2, the Cronbach’s alpha values ranged from 0.749 to 0.930 and the CR values ranged from 0.841 to 0.947. Furthermore, the AVE values ranged from 0.641 to 0.781. The internal reliability and convergent validity were suitable for this study, as indicated by the results. Examination of the square root of AVE and cross loading matrix was proposed by Fornell and Larcker [59], discriminant validity can be gauged. As seen in Table 3, the statistics highlighted in the corresponding correlation matrices are the square root of each latent construct’s AVEs. Compared to the total value of the inter-construct correlations, these values are higher. A convincing correlation with their own measures compared to other measures exists for all constructs, indicating that discriminant validity has been accomplished.

### 3.2. Hypotheses Testing

This study examined the factors affecting passengers’ intention to use the Spider Card for public transportation systems in Thailand. A bootstrap sampling procedure drawn from Efron and Tibshirani [60] with 388 cases was used to determine the significance of the path coefficients in the structural model. Table 4 shows a list of the structural estimates and hypotheses testing results. Six of eight hypotheses (H1, H3, H4, H6, H7, H8) were supported at the 0.05 level whereas the remaining two (H2 and H5) are not supported.

Figure 2 demonstrates the results of the standardised path coefficient for each hypothesis tested from the PLS-SEM analysis. The impartial variables of the full sample can be employed to explain 65.60 percent of the inconsistency for users’ intent to use the Spider Card. An assessment was carried out on the explanatory power of the model for each construct. The resulting *R*^2^ for perceived value and performance expectancy were 64.20 percent and 38.89 percent, respectively. The variations in the intention to use Spider Card can be used to clarify the variables in the research model.

## 4. Discussion and Implications

### 4.1. Discussion and Theoretical Implications

The findings in this study demonstrate that PE has a positive influence and is significant for the intention to use the Spider Card. It was found that people tend to use the Spider Card if they believe it is useful for their daily life and the fare payment scheme is more advantageous than using traditional or existing fare payment media. These findings are in line with previous smartcard and payment ticketing studies undertaken by Di Pietro, et al. [35], Loo, et al. [61] and Yeow and Loo [52], which found that PE is an imperative variable for predicting users’ intentions in the transportation context and also consistent with [13,19,33,62] which specified that PE is one of the dominant factors in technology intention and adoption aspects. This study also found that EE is the most significant factor affecting PE, but not with the behavioural intention to use the Spider Card. These findings might be due to the fact that most respondents basically travel with mass rapid transit services. For example, the BTS sky train uses the Rabbit Card (a store-value smartcard payment system), which is the same concept as the Spider Card. Thus, many people have already gained experience and familiarity with smartcard store-value tickets. Therefore, people might believe that there is no new technology that would make them feel that the common ticketing system is too complex or difficult to use. The same effect of EE associated through PE was also found in the work of Eyuboğlu and Sevim [54], which likewise indicated that perceived ease of use (e.g., EE) has a direct effect on perceived usefulness (e.g., PE) for the acceptance of contactless credit cards in Turkey. However, there is an inconsistency with Wu, et al. [37], who showed these variables have a significant and positive influence on behavioural intention to use the iPass smartcard for Taiwan’s MRT system. 

Regarding FC, this study found that FC is significantly related to behavioural intention. This indicates that using the Spider Card is reinforced by individual opinions about the resources, infrastructure and support available from ticketing operators and associated agencies. Loo, et al. [61,63] and Wu, et al. [37] also identified that FC has a positive effect on the intention to use multipurpose smartcard and electronic ticketing technology. During the validation of the theoretical model, it was found that SI was not validated as a significant predictor of the intention to use in the context of this study. The analysis of the SI also proved to be relevant and have a positive relationship in the prediction of behavioural intention to use technology [13,19,33,37,64]. Conversely, this research was consistent with a previous study by Eyuboğlu and Sevim [62], which found comparable results investigating contactless credit cards. 

In terms of the relationship between perceived value and behavioural intention, this finding was supported by the previous results in [16,43,45]. Sumaedi, et al. [45] likewise revealed similar results in terms of the impact of perceived value. Further, Wen, et al. [43] and Sumaedi, et al. [45] also established the impact of perceived sacrifice on perceived value as identified in this study. Since perceived convenience and perceived value are similarly increased with decreasing perceived sacrifices, for example, passengers have more convenience when transferring to another transit system, this boosts the intent to use the Spider Card while positively influencing its overall use. Thus, the cost to passengers for using common ticketing services should be taken into account by transit ticket operators for both monetary sacrifice and non-monetary sacrifice (e.g., time costs, search costs, convenience to top-up ticket). Last, it is obviously meaningful in terms of accessibility. The research results confirm that convenience is consistent with previous findings by Liu, et al. [49]. That study revealed a comparable connection between the influences of perceived convenience on perceived value. Transport users will be more willing to use the Spider Card if they believe it is more convenient than other means in terms of monetary benefits and time savings, as proven by the results. 

### 4.2. Practical Implications

The current study sheds light on the links between behavioural intention and aspects of technology adoption, as well as between behavioural intention to use and value perceptions by exploring the concrete components of perceived value and technology adoption in transportation smartcard ticketing systems. This research offers significant suggestions. First, perceived value, facilitating conditions and performance expectancy are significant factors for passengers’ intention to use the Spider Card. The results likewise show that public transit passengers are more likely to use the Spider Card if they view it as being more beneficial than other means. It can also be seen that the perceived value influence has a more significant impact than the UTAUT constructs, signifying that users are more amenable to the Spider Card based on the degree of value it can or may afford. The inference is that public transit passengers concern about the level of value they will gain. Simply put, people care about whether the benefits of using the system outweigh any potential disadvantages. 

The findings of this study have important implications for the Mass Rapid Transit Authority of Thailand (MRTA) who act as the Spider Card operator as well as being in charge of a central clearing house for the Spider Card ticketing system. MRTA could enhance passengers’ expectations by offering more advantageous perks and rewards, hence, transportation authorities and the MRTA should think about utilising effective marketing approaches. Moreover, MRTA should accelerate the process of allowing Spider Cards to be used for non-transit payment, such as in convenience stores or for bill payment services. Second, the factor of effort expectancy comprises the most significant factor involving performance expectancy, but not the behavioural intention to use the Spider Card. This also suggests that public transit passengers will be more motivated to use the Spider Card and perceive that it is more beneficial to them if they observe greater ease of use and more benefits compared to using other means. As suggested by the results, the MRTA should stress the ease and speed of using the Spider Card compared to traditional existing payment methods in transit systems, which will increase acceptance of the Spider Card usage. Further, it would be profitable if passengers knew more about the advantages and efficacy of integrated smartcard ticketing technology. Enhancing users’ acceptance of the Spider Card is necessary to increase the number of overall users. Third, this research also confirmed that facilitating conditions are influential factors affecting passengers’ intention to use the Spider Card. This shows that the MRTA should facilitate adequate payment as well as top-up ticket channels in various ways to facilitate passengers’ use of the Spider Card by considering online platforms or convenience stores so that it is more convenient for passengers. A far-reaching promotion to publicise the benefits of the common ticketing system and inspire the widespread use of the Spider Card should be coordinated by the transport authorities.

### 4.3. Limitations and Future Research

Despite the promise and usefulness of the findings in this study, various limitations were apparent. First, the research results obtained from the study may not be generalised to other countries as the data was collected only in Bangkok, Thailand. If the research model is operated in other countries, the findings may be different. Second, as the progress of integrated transit payment ticketing technology has only been launched in the capital city of Thailand, the results only reflected the residents of the capital city. If the Spider Card is used in other major cities of Thailand in the future, any subsequent research should consider using a larger sample with more diverse locations. Third, the data was collected from an online survey platform rather than a paper-based survey due to cost and time limitations as well as an inability to reach more groups of people who use public transportation in daily life. The results also revealed that most respondents knew about Spider Cards, but most of them did not have much experience using them. Therefore, users who have already adopted the system might have different perceptions. Finally, this study used cross-sectional data. However, understanding passengers’ intention to use the Spider Card should be a continuous process. Thus, longitudinal data would provide a better picture of this. A worthwhile pursuit for future study is to identify users’ continuous or discontinuous use of Spider Cards, also called a longitudinal study. The results of such research could enhance the value of this study. This study only examined the factors affecting the intention to use the Spider Card without examining the moderating effects of demographic variables. Future research should consider examining demographic variables such as age, gender, education level and personal income.

## 5. Conclusions

This study focused on passengers’ intention to use the common ticket locally known as the Spider Card in Thailand. Based on prior research, this study also serves as a bridge by extending the understanding of passengers’ behaviour in the public transportation context on technology acceptance in early adoption, as well as how perceived value can influence the intention to use the Spider Card. The results revealed that perceived value, performance expectancy and facilitating conditions have a positive influence on passengers’ intention to use the Spider Card. Besides, perceived value is the main determinant factor influencing behavioural intention. Perceived value is positively influenced by perceived convenience but negatively influenced by perceived sacrifice. Furthermore, effort expectancy is the most important factor affecting performance expectancy, but insignificant on behavioural intention to use the Spider Card. Nonetheless, this study identified that social influence was not deemed to be a substantial predictor of intention in the context of this study. Finally, this research indicated that the created model might appreciably impact comprehension of the factors that induce the utilisation of the Spider Card for public transportation systems in Thailand. The study’s findings may also be useful as a guideline for public transport authorities and ticket operators to comprehend passengers’ desires when utilising this specific type of technical service system.

## Figures and Tables

**Figure 1 behavsci-09-00046-f001:**
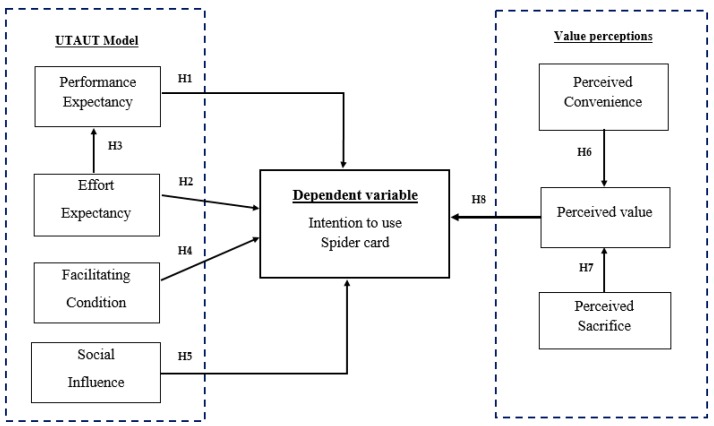
The proposed research model.

**Figure 2 behavsci-09-00046-f002:**
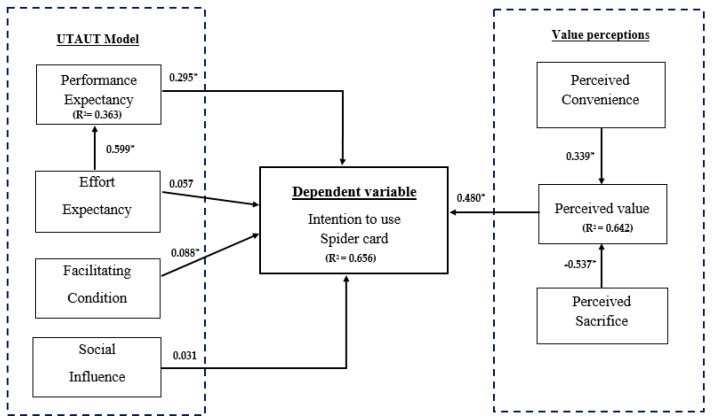
Standardised path coefficients. **(Note: *** indicate the significant at the *p* < 0.05 level.)

**Table 1 behavsci-09-00046-t001:** Demographic characteristic of the respondents.

Demographics	Items	Frequency	Percentage (%)
Gender	Male	115	29.6
	Female	273	70.4
Age	Below 17	1	0.26
	17–24	31	7.99
	25–35	211	54.38
	36–45	105	27.06
	46–55	24	6.19
	56–64	15	3.87
	Above 65	1	0.26
Education	Middle school	2	0.52
	High school	4	1.03
	Diploma	5	1.29
	Bachelor’s Degree	188	48.45
	Above Bachelor’s Degree	189	48.71
Occupation	Student	22	5.67
	Government Officer	134	34.54
	Company Employee	171	44.07
	Business Owner/Freelance	43	11.08
	Housewife	14	3.61
	Others	4	1.03
Personal income Baht per month (USD.)	Less than 10,000 Baht (312 USD.)	20	5.15
* Note 1 USD = 32 Baht	10,000–15,000 Baht (312–469 USD.)	26	6.70
	15,001–25,000 Baht (469–781 USD.)	106	27.32
	25,001–35,000 Baht (781–1093 USD.)	98	25.26
	35,001–50,000 Baht (1093–1562 USD.)	76	19.59
	More than 50,000 Baht (1562 USD.)	62	15.98
Usage experience	>0 and <3–5 times	379	97.68
(times per week)	≥3–5 times	9	2.32
Total = 388			

**Table 2 behavsci-09-00046-t002:** Results of all item loadings, reliability and convergent validity testing.

Construct	Items	Loadings	Cronbach’s Alpha	CR	AVE
Performance expectancy	PE1	0.863	0.872	0.912	0.722
	PE2	0.855			
	PE3	0.844			
	PE4	0.837			
Effort expectancy	EE1	0.865	0.816	0.891	0.732
	EE2	0.882			
	EE3	0.818			
Facilitating conditions	FC1	0.841	0.775	0.867	0.686
	FC2	0.784			
	FC3	0.857			
Social influence	SI1	0.907	0.789	0.845	0.647
	SI2	0.747			
	SI3	0.748			
Perceived convenience	PC1	0.824	0.880	0.917	0.735
	PC2	0.856			
	PC3	0.892			
	PC4	0.856			
Perceived sacrifice	PS1	0.724	0.749	0.857	0.667
	PS2	0.863			
	PS3	0.856			
Perceived value	PV1	0.817	0.891	0.924	0.754
	PV2	0.871			
	PV3	0.915			
	PV4	0.868			
Intention to use	IU1	0.890	0.930	0.947	0.781
	IU2	0.912			
	IU3	0.894			
	IU4	0.863			
	IU5	0.858			

**Notes:** CR = composite reliability; AVE = average variance extracted

**Table 3 behavsci-09-00046-t003:** Discriminant validity statistics of each construct.

Constructs	PE	EE	FC	SI	PC	PS	PV	IU	Mean (S.D.)
PE	**0.850**								
EE	0.603	**0.855**							
FC	0.479	0.477	**0.829**						
SI	0.384	0.393	0.495	**0.800**					
PC	0.751	0.599	0.565	0.430	**0.857**				
PS	−0.580	−0.497	−0.482	−0.464	−0.655	**0.817**			
PV	0.612	0.560	0.534	0.531	0.691	−0.759	**0.868**		
IU	0.676	0.558	0.527	0.465	0.772	−0.665	0.758	**0.884**	

**Note:** The average variance extracted (AVE) values for each latent construct are denoted by the underlined values on the diagonal, while the measures under the diagonal signify squared inter-construct correlations.

**Table 4 behavsci-09-00046-t004:** Structural estimates and hypotheses testing results.

Paths	Standard Coefficient (β)	*t*-Value	Result
H1: PE → IU	0.292	4.972	Supported
H2: EE → IU	0.059	1.328	Not supported
H3: EE → PE	0.536	9.279	Supported
H4: FC → IU	0.086	2.107	Supported
H5: SI → IU	0.032	0.814	Not supported
H6: PC → PV	0.340	7.130	Supported
H7: PS → PV	-0.536	11.696	Supported
H8: PV → IU	0.482	9.511	Supported

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
