# Peer review of "Factors Influencing the Intention to Use the Common Ticketing System (Spider Card) in Thailand"

_behavsci, 2019, doi:10.3390/bs9050046_

Reviewer 1 Report

The authors describe in the paper the model to assessing the factors influencing the intention to use the so called “Spider Card” in Thailand. The goal of the paper is clear, and it is well motivated. The article covers an adequate methodology to address  the research question. But please add the description why the sample size of at least 100 is suggested (according to Cohen’s rule)? Please add the description, because in my opinion the 408 respondents from Bangkok is too small.  The sentence “the data was collected only in Thailand” in the section “4.3. Limitations …“ is too wide -  the data was collected only in Bangkok not in whole country.

The section: "Practical Implications" must be  improved.

I am asking you to extend the discussion to the possibility of using the research results by public transport institutions - to what extent are the results obtained useful? How can they be used?

Author Response

Point 1: The authors describe in the paper the model to assessing the factors influencing the intention to use the so called “Spider Card” in Thailand. The goal of the paper is clear, and it is well motivated.

Response 1: We thank reviewer for generous comments on the manuscript. We appreciate very much and have edited the manuscript to address your concerns.

Point 2: The article covers an adequate methodology to address the research question. But please add the description why the sample size of at least 100 is suggested (according to Cohen’s rule)? Please add the description, because in my opinion the 408 respondents from Bangkok is too small.  

Response 2: We thank Reviewer to indicate these unclear points to our attention. In order to eliminate some misunderstanding point as well as make it clearer. The following sentences are modified on page 8. 

....the final dataset. According to Cohen’s rule [40], a robust effect size for the sample size should be at least 10 times the amount of the hypotheses. The research model examined 8 hypotheses; therefor, the sample size of at least 80 is suggested.

Point 3: The sentence “the data was collected only in Thailand” in the section “4.3. Limitations …“is too wide -  the data was collected only in Bangkok not in whole country.

Response 3: We thank Reviewer for this valuable suggestion. In the section 4.3 Limitations of the revised manuscripts in page 13. for the sentence “the data was collected only in Thailand” we have now changed to “the data was collected only in Bangkok, Thailand”.

Point 4: The section: "Practical Implications" must be improved.

Response 4: We thank reviewer for this helpful comment. We agree with reviewer that our Practical Implication section could be more improved. We have now revised the Practical Implication section in the revised manuscripts on page 12.

4.2. Practical Implications

The current study sheds light on the links between behavioural intention and technology adoption aspects, as well as between behavioural intention to use and value perceptions aspects by exploring the concrete components of perceived value and technology adoption in transportation smartcard ticketing systems. This research offers significant suggestions. First, perceived value, facilitating conditions and performance expectancy are significant factors for passengers’ intention to use the Spider Card. The results likewise show that public transit passengers are more likely to use the Spider Card if they view it as being more beneficial than other means. It can also be seen that the perceived value influence has a more significant impact than the UTAUT constructs, signifying that users are more amenable to the Spider Card based on the degree of value it can or may afford. The inference is that public transit passengers concern about the level of value they will gain. Simply put, people care about whether the benefits of using the system outweigh any potential disadvantages.

The finding of this study has important implications for Mass Rapid Transit Authority of Thailand (MRTA) whose act as the Spider Card operator as well as in charge of a central clearinghouse for the Spider Card ticketing system.MRTA could enhance passengers’ expectation by offering more advantageous perks and rewards, transportation authorities and MRTA should think about utilising effective marketing approaches. Moreover, MRTA should accelerate the process of allowing Spider Cards to be used for non-transit payment, such as in convenience stores or for bill payment services. Second, the factor of effort expectancy comprises the most significant factor involving performance expectancy, but not the behavioural intention to use the Spider Card. It is also pointed out that public transit passengers will be more motivated to use the Spider Card and perceive that it is more beneficial to them if they observe greater ease of use and more benefits compared to using other means. As suggested by the results, MRTA should stress the ease and speed of using the Spider Card compared to traditional existing payment methods in transit systems, which will increase acceptance of the Spider Card usage. Further, it would be profitable if passengers knew more about the advantages and efficacy of integrated smartcard ticketing technology. Enhancing users’ acceptance of the Spider Card is necessarily increasing the number of overall users. Third, this research also confirmed that facilitating conditions are influential factors affecting passengers’ intention to use the Spider Card. This shows that MRTA should facilitate adequate payment as well as top-up ticket channels in various ways to facilitate passengers’ use of the Spider Card by considering online platforms or convenience stores so that passengers can gain more convenience. A far-reaching promotion to publicise the benefits of the common ticketing system and inspire the widespread use of the Spider Card should be coordinated by the transport authorities.

Point 5: I am asking you to extend the discussion to the possibility of using the research results by public transport institutions - to what extent are the results obtained useful? How can they be used?

Response 5: We are grateful for this comment as it points to an important practical implication of this study. The objective of this study is to explain which variables may be relevant in this context. Understanding these factors can help the transport policy makers, such as The Ministry of Transport (MOT) and the Mass Rapid Transit Authority of Thailand (MRTA) should first investigate what is driving citizens to use this new smartcard ticketing services. This research can also be functional for Thailand transport authorities (MOT), encouraging the MRTA as the head of Spider Card operator to design a suitable smartcard payment ticketing solutions based on their passengers’ needs. For instance, this study can provide useful information regarding the main characteristics of all-in-one transportation smartcard. Spider card makes public transportation passengers more convenience, easy and faster access to public transport in those situations in which they can make a journey that involves transfers within or between different transport modes by using the same ticket.

Reviewer 2 Report

Theoretical framework must be improved and explained in more details: UTAUT and its dimensions should be more extensively analized (all dimension have to be defined) and explained, also concept of perceived vaule and its dimensions, specialy dimensions analized and included in research model. Also, research model and its hypothesis should be more connected to theoretical framework. Authors could also analized to impact of UTAUT dimension such as performance expectation to perceived value. Some theoretical researches concluded that customers expectations have impact to finel perceived value.

In Table 1. personal income should be presented in international currencies such as US dollar. 

Author Response

Point 1: Theoretical framework must be improved and explained in more details: UTAUT and its dimensions should be more extensively analized (all dimension have to be defined) and explained.

Response 1: We thank Reviewer for the valuable suggestion. We already explained in more detailed about the UTAUT and its dimensions and the following sentences are added on page 3. of the revised manuscript. and have cited some relevant references in your reply.

1.1.1. Unified Theory of Acceptance and Use of Technology (UTAUT) and its dimensions

    Several technology adoption theories have often been used for research studies in order to forecast personal different intention to adopt new technologies. A Unified Theory of Acceptance and Use of Technology (UTAUT) is one of the unified technology acceptance models formulated by Venkatesh, et al. [1], as a conceptual framework for understanding users’ intention and acceptance of technology in several contexts [2-3]. The UTAUT model integrated from an inclusive of eight dominant model of acceptance and use of technology theory including Theory of Reason Action (TRA) proposed by Fishbein and Ajzen [4], Technology Acceptance Model (TAM) proposed by Davis [5] and Davis, et al. [6] , Theory of Planned behavior (TPB) proposed by Ajzen [7], Combined TAM and TPB (C-TAM-TPB) proposed by Taylor and Todd [8], Motivation Model (MM) proposed by Vallerand [9], Model of PC Utilization (MPCU) proposed by Triandis [10] , Diffusion of Innovation Theory (DOI) proposed by Rogers [11] , Social Cognitive Theory (SCT) proposed by Bandura [12], which is connected all eight theories above as the unified theory between the main factors and latent variables. The key determinant constructs affected the intended use of new technology perspectives, which can be determined by three constructs including performance expectancy, effort expectancy and social influence. Performance expectancy is defined as the degree to which an individual believes that using the system will help them to accomplish gains in job performance[1], while effort expectancy can be defined as the degree of ease associated with consumers’ use of technology which is explains an individuals’ perception on the level of difficulty or ease of use an IT system [1] as well as a predictor for an individual’s intention to use of IT[13]. Social influence is the extent to which individuals perceives the degree of approval of a certain behavior by important referents (e.g., family and friends) believe he or she should use a particular technology. Moreover, Venkatesh, et al. [1] indicated that facilitating conditions and behavioral intention also have direct effect on usage behavior. Facilitating conditions defined as the degree to which an individual believes that an organizational and technical infrastructure exists to support use of the system[2]. Also, there are four individual characteristics including gender, age, experience, and voluntariness of use as moderators of UTAUT model[1] , However, the moderating variables were not mentioned in this study.

    The UTAUT has been used to explain behavioral intention and used in various context of information technology such as e/m-government services [33-34], e/m-payment [11, 19, 35-36], mobile technology [20], and transportation systems [13, 37]. Previous studies have applied the UTAUT model to explore the context of transit fare payment technology adoption, as seen in Wu, et al. [14] adopted the UTAUT model to investigate citizens’ acceptance and use of iPass transit smartcard in Taiwan’s Kaohsiung MRT system. The results showed that most constructs of the UTAUT have a strong positive influence on behavioural intention. This study also takes facilitating conditions as one of the determinant factors affecting the intention to use the Spider card. Previous studies have proven that facilitating conditions have a positive influence on technology adoption and intention [2].

Point 2: Concept of perceived value and its dimensions, specially dimensions analized and included in research model.

Response 2: We thank Reviewer for this suggestion. We now have added more in detail on the concept of perceived value and its dimensions in Section 1.1.2. – 1.2.4. of the revised manuscript on page 3. and have cited some relevant references in your reply.

1.1.2. The Role of Perceived Value

Perceived value is the major element of any successful marketing strategies. As first offered by Zeithaml [15], perceived value refers to an overall assessment of a product or service from the consumers based on perceptions of what is received and what is given.  Numerous scholars have extended their understanding of perceived value is the ratio or trade-off between benefit and cost, e.g. [16-19], which is a value-for-money conception. In general, perceived value can be linked to the value that consumers believe they will get from buying certain products and services. This is done by comparing with other similar products and services in terms of the financial and non-financial aspects of purchased products and services [15, 20]. Zeithaml [15] argued that various people view value as meaning a low price, while others see it as relative equality between the quality and price of a purchase. Another researcher, Lovelock [20] also suggested that perceived value may be considered a compromise between perceived benefits and perceived cost. Additionally, perceived value will probably by high when the perceived benefit is greater than any perceived cost. The opposite is also true [21].

A commonly accepted influence factor on behavioural intention and adoption is perceived value. A number of studies [19, 21-23] revealed that perceived value could be impacted by various precursor variables. Focusing on the transportation context, Lai and Chen [22] examined the influence of perceived value on behavioural intentions for major transport facilities. Apparently, perceived value has a direct influence on behavioural intentions, as found in their study. Sumaedi, et al. [24] likewise identified the importance of perceived value for the intention of users of public transit in Jakarta, Indonesia. Thus, it could be assumed that perceived value is a significant element of behavioural intentions to use common ticketing among passengers. Thus, the intention to use a common ticketing system will be high if passengers perceive high value.

1.13. Perceived Convenience

            Convenience is typically considered and debated in marketing and consumer behaviour literature [25]. Convenience is also recognised as an aspect of consumers’ attitudes in terms of the time and effort needed for the buying of specific goods and services[25]. Yoon and Kim [26]described perceived convenience as the degree of convenience individuals realise when using services to complete a task measured by time, location and accomplishment. Moreover, Colwell, et al. [27]likewise revealed that service convenience can be classified as a way to create added value for consumers by reducing the time and effort that they have to spend for products and services. When a product or service saves time for a consumer, it is deemed convenient. Conversely, a product or service is viewed as being convenient if it decreases the mental, emotive and physical drain felt by consumers. Based on the perspective provided by [25-26], in the context of transport services in this research, perceived convenience was defined as the degree of passengers’ recognition of convenience due to using the Spider Card as measured by less travel time and ease of inter-modality transport in addition to time savings in comparison with conventional pay systems. Further, experiential research by Liu, et al. [28] likewise showed that perceived convenience has an affirmative impact on perceived value toward the behavioural intention of using mobile coupon applications.

1.1.4. Perceived Sacrifice

Perceived Sacrifice typically concerns what must be surrendered or paid in order to gain certain goods or services [15]. As seen in literature from previous studies, perceived sacrifice may be separated into monetary and non-monetary, such as time, search, and psychological costs [15, 21, 29]. A sacrifice is obvious if a consumer must use time and travel some distance to buy a product or service. Moreover, on the basis of Zeithaml (1988)’s work, perceived sacrifice is often revealed as an indicator of the perceived value. In previous works relating to transportation, an immediate and adverse bond between perceived sacrifice and perceived value is verifiable. For example, the study conducted by Wen, et al. [21] revealed that perceived sacrifice is noteworthy according to statistics, meaning it has a direct unfavourable impact on perceived value. This indicates that low perceived value could be a consequence of high perceived sacrifice.  Quite the opposite, low perceived sacrifice could bring about high perceived value.

Point 3: Research model and its hypothesis should be more connected to theoretical framework.

Response 3: We agree with reviewer that hypothesis should be more connected to theoretical framework. Therefore, we have added The following sentences for the reader to clarify our research model and its hypotheses in section 1.3. Proposed Research model of our revised manuscript on page 6.

1.3 Proposed Research Model

    On the basis of the literature review and hypotheses presented above, the proposed research model is presented in Figure 1. This model illustrates the certain hypothesized relationship among the intention to use, the construct of UTAUT model and the value perceptions aspect.

Point 4: Authors could also analyze to impact of UTAUT dimension such as performance expectation to perceived value.

Response 4: We thank reviewer for this interesting suggestion and it would be providing an additional contribution for us in the next research step. Based on the existing literature reviews, our study has not yet been addressed on the impact of UTAUT dimension to perceived value. However, this issue would be providing some additional contribution and it should be considered for us in the next research step.

Point 5: Some theoretical researches concluded that customer expectations have impact to final perceived value.

Response 5: We thank reviewer for sharing this knowledge about the relationship between customer expectations and perceived value. However, based on our existing literature, the hypotheses of this study has not yet been considered at this point, but we will consider following your information in our next research step.

Point 6: In Table 1. personal income should be presented in international currencies such as US dollar.

Response 6: We agree with reviewer that personal income in Table 1. (Page 8) should be presented in international currencies. We now have added the US. Dollar currency in parentheses after Baht-Thai on Table 1.: Demographic characteristic of the respondents. in column Personal income (per month).

Round  2

Reviewer 2 Report

I do not requere additional corrections.